

# A new zenith looking narrow-band radiometer based system (ZEN) for dust Aerosol Optical Depth monitoring

A. Fernando Almansa[1,2,4], Emilio Cuevas[1], Benjamín Torres[3], África Barreto[1,2], Rosa D. García[1,5], Victoria E. Cachorro[4], Ángel M. de Frutos[4], César López[6], and Ramón Ramos[1]

[1]Izaña Atmospheric Research Center (IARC), Meteorological State Agency of Spain (AEMET), Santa Cruz de Tenerife, 38001, Spain
[2]Cimel Electronique, Paris, 75011, France
[3]Laboratoire d'Optique Amosphérique – UMR8518, Université des Sciences et Technologies de Lille, Villeneuve d'Ascq, France
[4]Group of Atmospheric Optics, University of Valladolid, Valladolid, 47011, Spain
[5]Air Liquide España, Delegación Canarias, Candelaria, 38509, Spain
[6]Sieltec Canarias S.L., La Laguna, 38230, Spain

*Correspondence to:* A. F. Almansa (fernando@goa.uva.es)

**Abstract.** A new zenith looking narrow-band radiometer based system (ZEN), conceived for dust aerosol optical depth (AOD) monitoring, is presented in this paper. The ZEN system comprises a new radiometer (ZEN-R41) and a methodology for AOD retrieval (ZEN-LUT). ZEN-R41 has been designed to be stand-alone and without moving parts, making ZEN-R41 a low-cost and robust instrument with low maintenance, appropriated to be deployed in remote and unpopulated desert areas. The ZEN-LUT method is based on the comparison of the measured Zenith Sky Radiance (ZSR) with a lookup table (LUT) of computed ZSRs. The LUT is generated with the LibRadtran radiative transfer code. The sensitivity study proved that the ZEN-LUT method is appropriated to infer AOD from ZSR measurements. The validation of the ZEN-LUT technique was performed using data from AErosol RObotic NETwork (AERONET) Cimel Electronique 318 photometers (CE318). A comparison between AOD obtained by applying the ZEN-LUT method on ZSRs (inferred from CE318 diffuse sky measurements) and AOD provided by AERONET (derived from CE318 direct sun measurements) was carried out at three sites characterized by a regular presence of desert mineral dust aerosols: Izaña and Santa Cruz in the Canary Islands, and Tamanrasset in Algeria. The results show a $R^2$ ranging from 0.99 at Santa Cruz to 0.95 at Tamanrasset, and a maximum root mean square error (RMSE) ranging from 0.010 at Izaña to 0.035 at Tamanrasset. The comparison of ZSR values from ZEN-R41 and the CE318 showed absolute relative mean bias (RMB) <10%. ZEN-R41 AOD values inferred from ZEN-LUT methodology were compared with AOD provided by AERONET, showing a fairly good agreement in all wavelengths, with absolute AOD differences < 0.030 and coefficient of determination ($R^2$) higher than 0.97.

## 1 Introduction

Atmospheric aerosols play an important role in the environment, especially affecting air quality and climate. Concerning the Earth's climate, atmospheric aerosols are one of the main drivers of climate change and the most uncertain of them (Stocker





et al., 2013). These atmospheric constituents can affect the Earth's radiative balance by scattering and absorbing the incoming solar radiation and the outgoing terrestrial radiation (aerosol-radiation interactions) but also by influencing cloud formation and reflectivity (aerosol-cloud interactions) (Klein et al., 2010; Hoose and Möhler, 2012; Stocker et al., 2013). Furthermore, aerosols are not uniformly distributed in the atmosphere showing a high spatial and temporal variability. For these reasons, a

good knowledge of the microphysical and optical properties of atmospheric aerosols is necessary all over the world. In this respect, aerosols remote sensing by means of satellite and ground-based devices is normally used to routinely monitor the aerosols columnar optical and microphysical properties, the latter obtained by inversion techniques (i.e., Dubovik and King (2000)). Regarding the aerosol optical properties, the Aerosol Optical Depth (AOD) and its spectral dependence are the most common and important parameters for aerosol characterization, widely used in models and satellite sensors.

Remote sensing from satellite platforms has proved to be an effective tool for global and long-term monitoring of aerosols in the atmosphere. Early satellite sensors, such as the Advanced Very High Resolution Radiometer (NOAA/AVHRR) (Stowe et al., 1997) or TOMS (Total Ozone mapping Spectrometer) (Herman et al., 1997; Torres et al., 1998), permitted the AOD estimation spanning from 1979 to the present. In recent years, a more advanced generation of active and passive satellite sensors for aerosol detection have been used to estimate aerosols radiative forcing on a global scale: such as MODIS -Moderate

Resolution Imaging Spectrometer-, MISR -Multi-angle Imaging Spectro-Radiometer-, both in operation since 2000 (Tanré et al., 1997; Zhang and Reid, 2006, 2010) or CALIPSO (Cloud-Aerosol Lidar and Infrared Pathfinder Satellite Observations) (Winker et al., 2010). However, as Li et al. (2009) suggested, satellite AOD retrievals are subjected to important uncertainties due to radiometric calibration, a priori assumed aerosol properties, cloud contamination and correction of the surface effect. In particular, a reliable determination of the ground reflectance is mandatory, because the signal measured by the on-board

sensor is the sum of the top of atmosphere (TOA) reflection and the surface reflection. This reliable determination is especially important over bright land surfaces like deserts, where the main dust sources are located. Moreover, the temporal resolution of satellite borne sensors over a specific point is quite limited (one observation per day as maximum) which represents a high limitation for dust storms monitoring since they develop very fast in time periods of few hours.

There are currently extensive networks of ground-based sun-photometers all over the world dedicated to aerosols monitor-

ing. These networks are integrated by very precise instruments, close to the 0.02 AOD accuracy suggested as a goal by the World Meteorological Organization (WMO) (Schmid et al., 1999; WMO, 1993). The AErosol RObotic NETwork (AERONET) network (Holben et al., 1998) is the most widespread network for aerosol monitoring. AERONET uses the Cimel Electronique 318 photometer as standard instrument (hereinafter CE318). At present, this network provides unique long-term and aerosol optical, microphysical and radiative properties open-access data what constitutes an important source of information for cli-

mate and environmental sciences, since their results have been widely tested under quite different conditions (Eck et al., 1999, 2001, 2003a, b, 2008, 2009, 2010; Holben et al., 2001). However, there is a lack of stations in desert regions in the Northern Hemisphere, which are the most important source of mineral dust, leading to an unsatisfactory description of dust cycles. Although AERONET sun-photometers operate automatically, they need to be checked on a daily basis and frequently maintained by trained staff to assure data quality. These instruments operate using a sun tracker, which is a moving element and the origin

of most of the operational problems. In addition, the use of this tracking system considerably increases the power consumption,



which constitutes a crucial aspect in many remote sites not connected to national or regional electric grids. The relatively high cost of sun-photometers is a huge constraint for many developing countries located in desert regions.

For all the above reasons, we present in this paper a new system (ZEN), composed of the radiometer ZEN-R41 and a new methodology especially conceived to estimate AOD from downwelling Zenith Sky Radiance (ZSR) observations for desert dust
conditions. This system has been jointly developed by SIELTEC Canarias S.L. company (SIELTEC) and the Izaña Atmospheric Research Center (IARC) from the State Meteorological Agency of Spain (AEMET). The use of ZSR as the direct measurable magnitude simplifies the design of the radiometer ZEN-R41, avoiding the use of sun tracker, making it more robust, automated and available at lower cost than classical sun-photometers. However, the AOD calculation is not as straightforward and precise as using direct sun measurements, and some a priori assumptions must be made about aerosol properties, and significant
additional efforts must be done in the modeling part. As made in the techniques employed by some on-board satellites sensors (Kaufman et al., 1997; Tanré et al., 1997; Stowe et al., 1997; Torres et al., 1998), we have developed a look-up table (LUT) based methodology (ZEN-LUT) specifically designed for desert aerosols to estimate AOD. The LUTs are composed by a set of simulated ZSR and AOD values obtained with the radiative transfer code LibRadtran (Mayer and Kylling, 2005). The AOD is then inferred by minimizing a function which depends on the difference between simulated and measured ZSR at all used
wavelengths. In this case, the surface albedo does not have as much influence as in the upwelling radiance measured by satellite sensors. These characteristics make the ZEN a suitable system for aerosol monitoring in remote desert areas, filling the current observational gaps and being also useful in the validation of satellite sensors and models dust products.

The test sites and ancillary observations used in this study are described in Section 2. In Section 3 a description of the new ZEN-R41 radiometer and a performance comparison with CE318 are given. The AOD retrieval method, including the
results of a sensitivity study performed to examine the impact of key input parameters as well as to assess the influence of the instrumental errors, is examined in Section 4. Section 5 describes the AOD comparisons between ZEN-R41 and collocated CE318. Finally, the main conclusions of this work are presented in Section 6.

## 2   TEST SITES AND ANCILLARY INFORMATION

### 2.1   Test Sites

The ZEN-LUT methodology and the ZEN-R41 instrument have been tested in three different sites impacted by the presence of desert mineral dust (Fig. 1).

The Izaña station (Izaña, The Canary Islands, Spain; 28.3°N, 16.5°W, 2373 m a.s.l.) is a high mountain subtropical station representative most of the time of atmospheric background conditions as a consequence of its location over a strong temperature inversion layer result of the combined effect of general subsidence associated with the descending branch of the Hadley
cell and the presence of cool trade winds in lower levels. However, the proximity with the Saharan desert introduces an important influence of mineral dust on its aerosols climatology. As Basart et al. (2009) showed, there is an enhancement of dust transport from the Sahara at Izaña altitudes during summer ($AOD_{675nm} > 0.15$ and large particles with Angström Exponent, $\alpha_{440-870nm} < 0.25$), with AOD values $< 0.15$ prevailing in the rest of seasons, especially in winter, which represent $\sim$





85% of the overall conditions. Guirado (2014) also performed a detailed aerosol characterization at Izaña. This author found predominant dust conditions are associated to $AOD_{500nm} > 0.10$ and large particles with $\alpha < 0.60$. Izaña Observatory is part of the World Meteorological Organization (WMO) Global Atmosphere Watch Programme (GAW) and the Network for the Detection of Atmospheric Composition Change (NDACC). Izaña is an absolute sun calibration site of AERONET and a

World Radiation Center (WRC) Global Atmospheric Watch Precision Filter Radiometer (GAW/PFR) station and calibration site. The Izaña station is also a WMO CIMO (Commission for Instruments and Methods of Observation) Testbed for Aerosols and Water Vapour Remote Sensing Instrument (WMO, 2014).

Santa Cruz de Tenerife station (SCO, The Canary Islands, Spain; 28.5°N, 16.2°W; 52 m a.s.l.) is an urban station located at sea level. The aerosol climatology at this station is dominated by the well-mixed combination of fine fraction of pollution

aerosols and coarse model marine particles (prevailing $AOD_{675nm} > 0.15$) with mineral dust influence from spring to autumn (Basart et al., 2009), increasing AOD and reducing $\alpha$ values. Following the work developed by Guirado et al. (2014), those situations with AOD > 0.15 and $\alpha < 0.5$ can be considered as prevalent dust conditions at this station.

Tamanrasset station (TAM, Algeria; 22.8° N, 5.5° E, 1377 m a.s.l.) is located in southern Algeria in a key location near the most important dust sources of Mali, Argelia, Lybia and Chad, with little impact of industrial activities. Guirado et al.

(2014) performed a thorough study to characterize aerosols in this Saharan station. They found that desert mineral dust is the predominant aerosol type in this station, where the dry-cool (winter) season is characterized by prevailing clear sky conditions ($AOD_{440nm} \sim 0.09$ and $\alpha \sim 0.62$), while high turbidity events with coarse dust particles are frequent during the wet-hot (summer) season, associated to an $AOD_{440nm}$ modal value of 0.15 and $\alpha \sim 0.4$.

## 2.2 Ancillary information

### 2.2.1 CE318 sun-photometer and AERONET network

We have used in this work AOD data provided by AERONET (Holben et al., 1998) to validate the results obtained with the ZEN-R41 radiometer and the ZEN-LUT technique. AERONET also provides information about microphysical and optical parameters, such as particle size distribution, refractive indices, single scattering albedo (SSA) or phase function using the inversion algorithm developed by Dubovik and King (2000).

The standard instrument used in AERONET is the CE318 sun-photometer, which performs direct sun and diffuse sky measurements. Direct sun measurements performed at 340, 380, 440, 500, 675, 870, 940, and 1020 nm are used to derive accurate AOD and precipitable water vapor with typical AOD uncertainties for field instruments ranging between ±0.01 and ±0.02, with the higher errors in the UV spectral range (Eck et al., 1999). Diffuse sky measurements with two different routines are performed to infer the aerosol optical and microphysical properties: the almucantar (ALM) and the principal plane (PPL).

In the ALM routine, the azimuth angle is varied while the zenith angle is kept constant (equals to the solar zenith angle). On the other hand, in the PPL routine, the zenith is varied while the azimuth angle is kept constant (equals to the solar azimuth angle). The ALM measurements are performed in two wings: right (azimuth angle displaced towards the right of sun position) and left (azimuth angle displaced towards the left of sun position) wings. In a homogeneous atmosphere, the signal measured at



both wings should be equal, so this fact can be used to detect wrong data, such cloud contaminated measurements. Contrary to ALM, PPL is not symmetric and the cloud screening is not evident. This is one of the reasons why AERONET does not use PPL for the inversion product retrievals (more details in Torres et al. (2014)). However, the PPL measurements are very important in our approximation since the radiance value in the zenith can be interpolated. This is not possible with ALM measurements.

In the present study, the CE318 ZSR values are obtained by interpolating the PPL measurements to the zenith position, so we cannot perform a cloud screening using only these measurements. Therefore, we have used all-sky camera images to remove cloud contaminated ZSR data, if available.

### 2.2.2   SONA all sky camera

In the present study we have checked clouds interference by means of independent measurements from all-sky Automatic
Cloud Observation System SONA cameras developed by SIELTEC (González et al., 2012). The SONA cameras have been used to detect and remove CE318 and ZEN-R41 cloud-contaminated ZSR data at Izaña and Santa Cruz stations. Cloud cover detection was performed by analyzing each individual hemispheric image from the total sky cameras, identifying the presence of visible clouds around the zenith.

## 3   THE ZEN-R41 NARROW-BAND RADIOMETER

The ZEN-R41 is a prototype radiometer jointly developed by SIELTEC and IARC designed to obtain AOD from downwelling zenith sky radiation at different wavelengths. This prototype incorporates collimating lenses and internal baffles to achieve a $\sim 3°$ field of view. ZEN-R41 is equipped with four Silicon detectors (350-1100nm) and four optical filters of 10 nm FWHM with nominal wavelengths centered in 440, 500, 675 and 870 nm, respectively. These filters are hard coated to prevent ageing of their optical properties. The measurements, made simultaneously in the four channels, are amplified and acquired with
16bit resolution (65.536 counts per each level of amplification). Inside the instrument there are sensors for internal humidity and temperature monitoring, as well as a fan for temperature homogenization and electronic components protection. Using this internal information, ZEN-R41 signal is temperature-corrected, allowing to minimize the temperature dependence of the silicon detectors. It is equipped with an aluminum weatherproof (IP67 grade) small case (white powder coated), with no moving parts and protected by a thick borosilicate bk7 window (see Fig. 2), preventing from damages such as scratches or direct impacts.
The device has also a plate base which allows leveling and fixing the instrument on flat and rigid surface. As a result, the instrument's design is very robust and operates in a wide temperature range, between -40°C and 85°C. It is possible to include a blower in the external case to facilitate removal of dirt, dust or any other element that can affect the measurement. In case of remote locations, it is possible to use solar panels as power source.

The on-board electronics comprises an internal 16-bits datalogger, and a 4MB internal memory with Ethernet communica-
tions for data acquisition, display, download, setup and diagnose of the instrument. On-board processing is possible through its microcontroller board, with 16 MIPS of CPU speed, and 96KB of RAM memory, which result in a fast user-oriented data





delivery (no additional parts or components are required) and a friendly-user interface. The power and communications are performed together through a PoE port.

The instrument can work with fixed IP or in DHCP mode. Data can be downloaded manually or automatically, and also can be sent via UDP or requested via TCP / IP protocol. It makes ZEN-R41 an instrument that might form part of a ZEN-1 network

for aerosol monitoring. This instrument can be also used independently, or as a dependent sensor part of a weather station or other instruments which require AOD as input.

The web interface shows alarms when these internal variables are out of the normal ranges. All the settings and calibration factors can be entered via a web graphical user interface.

ZEN-R41 was calibrated using an integrating sphere at the IARC facilities in order to convert the output signal into radiance

units ($W/m^2/sr/nm$). We performed a set of 10 measurements of the sphere's radiance, in which we observed a very low variability in ZEN-R41 digital counts ($\sim$2‰). Since the uncertainty involved in this calibration procedure was established in $\sim$ 5% by Walker et al. (1991) and assuming the sphere has a perfect precision (Holben et al., 1998), we can use this information to estimate the ZEN-R41 radiance uncertainty in a $\sim$ 5%.

### 3.1   ZEN-R41 and CE318 radiometric comparison at Izaña

In order to check the goodness of the radiometric measurements of the ZEN-R41, we have compared ZSR observations from this device with those provided by CE318 instruments, derived from PPL measurements. The ZSR measurements performed in Izaña station in 2015 were cloud screened making use of ancillary SONA all-sky images. In this comparison we have not included ZSR data for SZA $< 20°$ and SZA $> 65°$ as we have detected higher discrepancies in ZSR for such SZA ranges. In the case of SZA $< 20°$, the signal measured by ZEN-R41 device is larger than that obtained with CE318, which might be attributed

to a lower stray light rejection and/or a larger field of view of ZEN-R41 radiometer. For SZA $> 65°$, we attribute the observed discrepancies to lower signals measured, which reduce the signal to noise ratio, increasing the instrument's uncertainty. We present in Table 1 a basic statistics of ZSR intercomparison for the four coincident spectral bands (440, 500, 675 and 870 nm). ZSR measured by both instruments are highly correlated, with coefficients of determination (R) $\sim$ 0.99 for the four spectral ranges, discarding possible nonlinearities of ZEN-R41 in the selected measurement range. However, the relative mean bias

results (RMB) showed that ZEN-R41 ZSR slightly overestimates at 675, 500 and 440 nm, but underestimates at 870 nm. RMB also showed a moderate variability in the measured ZSR, with values ranging from -6.3% at 870 nm to 9.2% at 500 nm, slightly higher than the 5% calibration uncertainty value. Possible explanation for such results might be attributed to other types of instrumental errors and constraints, such as leveling, or stray light effect.

### 4   ZEN-LUT METHOD

For estimating the AOD from downwelling ZSR measurements under cloud-free conditions, we used the ZEN-LUT method. This method is based on the comparison of measured ZSRs, corrected from earth-sun distance, with a set of simulated ZSR values at the four wavelengths available in the ZEN-R41 radiometer (440, 500, 675 and 870 nm). The LUT was determined





using the LibRadtran radiative transfer model (RTM). This model is available in a complete software package containing a suite of tools for radiative transfer calculations in the Earth's atmosphere (freely available from http://www.libradtran.org) (Mayer and Kylling, 2005; Emde et al., 2016). The LibRadtran structures the atmosphere as multi-layers, considering the vertical profiles of temperature, pressure, and atmospheric components, such as gases and aerosols. A complete treatment

of the absorption scattering processes offers hundreds of options and input parameters to handle all the structure that has a detailed RTM. It also includes several libraries which help to describe the atmosphere and the ground surface contribution on the simulated radiation field. Concerning the aerosol contribution, we have used the OPAC (Optical Properties of Aerosols and Clouds) library (Hess et al., 1998). The aerosols in OPAC can be defined through 10 basic components. These are: water insoluble (INSO), water soluble (WASO), soot (SOOT), two sea salt components (sea salt accumulation mode or SSAM and

sea salt coarse mode or SSCM), four mineral dust components (mineral nucleus mode –MINM-, mineral accumulation mode –MIAM-, mineral coarse mode –MICM- and mineral transported –MITR-), and the sulfate component (SUSO). Additionally, LibRadtran includes four spheroids components (MINM, MIAM, MITR, and MICM spheroids) to define the mineral dust aerosols. The effects of relative humidity are taken into account for those components affected by water. Every component is defined by their microphysical properties, that is, the refractive index, mostly taken from d'Almeida et al. (1991), and a

log-normal size distribution (Deepak and Gerber, 1983). Then, through Mie theory or Tmatrix method in the case of spheroids, the optical properties are calculated for every component and normalized to 1 $particle/cm^3$.

The ZSR values were calculated by using the radiative transfer equation (RTE) solver CDISORT (Buras et al., 2011). The extraterrestrial solar flux was selected from Kurucz (1992) with a spectral resolution of 1 nm. The mid-latitude summer standard model for the atmosphere profile and the molecular absorption, parameterized with the LOWTRAN band model

(Pierluissi and Peng, 1985), as adopted from the SBDART code (Ricchiazzi et al., 1998), were used. We have also used a common normalized Gaussian filter function for the four considered spectral bands centered in the nominal wavelength (440, 500, 675 and 870 nm). To set the surface reflectance contribution, we have considered the albedo definitions given in the IGBP (International Geosphere Biosphere Programme) library, which originates from the NASA CERES/SARB Surface Properties Project (Belward et al., 1996).

Concerning the solar position, only the solar zenith angle (SZA) is given as input to the model, since the computed radiance in the zenith direction for a cloud free sky is invariable with the solar azimuth angle. The earth-sun distance correction is directly applied in the measured ZSR, so the ZEN-LUT ZSRs are only simulated for the mean earth-sun distance.

The aerosol optical properties were obtained with the OPAC library (Hess et al., 1998) by defining the height profiles of every aerosol component present at every layer. The height of all layers and the mix of aerosol components present at every

layer, except the boundary layer, were set up following the indications given in Hess et al. (1998). In the case of the boundary layer, we decided to set the mix of aerosol components dynamically, adopting the suggestions given in the Global Aerosol Data Set (GADS) report (Koepke et al., 1997) for desert regions. They proposed a mix of four different components, three mineral dust components (MINM, MIAM and MICM), plus a certain fix quantity of water soluble component (WASO). The ratio of three mineral dust components present in the mixture is variable and depends on the total mineral dust particle density





$(N_{mineral})$. The three mineral dust components are related with $N_{mineral}$ through the following expressions (Koepke et al., 1997):

$$ln(N_{MINM}) \sim 0.104 + 0.963 \cdot ln(N_{mineral}) \tag{1}$$

$$ln(N_{MIAM}) \sim -3.94 + 1.29 \cdot ln(N_{mineral}) \tag{2}$$

$$ln(N_{MICM}) \sim -13.7 + 2.06 \cdot ln(N_{mineral}) \tag{3}$$

In our calculations we have used the spheroids LibRadtran definitions.

It is important to note that the empirical relationships presented on these equations were derived for average conditions in desert areas, therefore they seem to be appropriated to describe the actual aerosols in Tamanrasset station, located in the middle of the Sahara desert, but not for Izaña and Santa Cruz Stations, which are located hundred kilometers away from the Sahara desert, over the North Atlantic. In the latter, the presence of large mineral dust particles is considerably reduced due to a faster deposition during its transport out of dust sources regions, and to the predominance of other aerosols in absence of Saharan intrusions. Because there is not a comprehensive definition of aerosols in terms of OPAC components, we propose a different mixture of components for these sites. This mixture involves the same WASO and MINM contribution as desert type aerosols, but the MIAM component is replaced by MITR, keeping the same relationship with $N_{mineral}$ as MIAM, and the MICM contribution is discarded.

We have generated the LUT using a set of $n$ values for $N_{mineral}$, ranging from 0 $particles/cm^3$ to 4000 $particles/cm^3$, in order to calculate $n$ height profiles, each one composed by all aerosol components present in every layer ($N_i(h)$). Then, with these profiles, we simulate a set of $n$ ZSR values for each SZA, in addition to the corresponding AOD in the ZEN-LUT.

Finally, to find out the right value for $N_{mineral}$ and thus the right AOD from the LUT, we have applied an adapted version of the method described in Tanré et al. (1997), originally designed to select the right aerosol model from satellites radiance measurements. In this case, the selection is performed by finding the minimum value of the following expression:

$$\epsilon_l = \sqrt{\frac{1}{N_\lambda} \sum_{\lambda=1}^{N_\lambda} \left( \frac{L_\lambda^m (\theta_v = 0, \theta_s) - L_{\lambda,l}^c (\theta_v = 0, \theta_s)}{L_\lambda^m (\theta_v = 0, \theta_s)} \right)^2} \tag{4}$$

where $L_\lambda^m$ and $L_{\lambda,l}^c$ are the measured (earth-sun distance corrected) and the computed radiances in wavelength $\lambda$, $N_\lambda$ is the total number of wavelengths and $l$ is the index indicating a value defined in the array $N_{mineral}$. $\theta_v$ is the viewing zenith angle which is equal zero and $\theta_s$ the solar zenith angle. Then the value of the index $l$ which minimizes the quantity $\epsilon_l$, indicates the right value for $N_{mineral}$ and AOD.





## 4.1 ZEN-LUT METHOD SENSITIVITY STUDY

The precision and accuracy of the solution given by the presented ZEN-LUT method are related with random and systematic errors. In the present work we have focused on performing a sensitivity study on the main systematic errors, which can be classified in two categories: instrumental errors, and errors made in the a priori considered RTM inputs, causing underestimation or

overestimation in the measured and simulated ZSR, which affects the retrieved AOD. In the first category, radiance calibration error, leveling error, misalignments of the optical parts, the effect of the finite field of view or the stray light contribution are included. The assumed value for surface albedo and the a priori considered mixture of aerosol components are included in the second category.

We have performed this sensitivity study using the inputs previously defined in Sect. 4.

This type of analysis is commonly used to identify the dominant contributors to the output variability. However, if the effect of each source of error is conveniently quantified, it is possible to estimate the contribution of each parameter to the uncertainty in the final AOD retrieval.

### 4.1.1 Instrumental error sensitivity

Taking into account previous studies in the literature (Eck et al., 1999; Basart et al., 2009; Guirado et al., 2014; Cesnulyte et

al., 2014; Cuevas et al., 2015) and AERONET climatology tables (http://aeronet.gsfc.nasa.gov/new_web/V2/climo_new), AOD desert stations such as Tamanrasset (Algeria), Ouarzazate (Morocco), Dakar (Senegal), and Solar Village (Saudi Arabia) show monthly and seasonal values typically ranging between 0.04 (Ouarzazate in January) and 0.67 (Dakar in June), with sporadic AOD maxima $> 1$ associated with strong desert dust outbreaks or local dust resuspension. Therefore, we have focused our sensitivity study in AOD conditions varying between 0.5, which corresponds to hazy conditions by dust, and 1, indicating

strong dust intrusions.

We have assumed the ZEN-R41 calibration uncertainty in a $\sim 5\%$ considering the comparison analysis with an integrating sphere presented in Sect. 3. In order to evaluate the influence of the overall instrumental errors on the inferred AOD we have perturbed the computed ZSR $\pm 5\%$ and $\pm 10\%$ for two aerosol loads (AOD $\sim 0.5$ and 1.0) at all the available wavelengths and SZAs ranging from 20° to 65° in 10° steps. Although this perturbed range should be statistically estimated by comparing

several ZEN-R41 instruments, we have selected these values according to the relative mean bias results obtained in the ZSR comparison between CE318 and ZEN-R41 presented in Sect. 3.1 and Table 1, in which we considered the CE318 as the reference instrument. Differences in AOD obtained from unperturbed and perturbed ZSR values are presented in Fig. 3 for 440 and 870 nm wavelengths. In case of AOD $\sim 0.5$, we obtained absolute differences $< 0.05$ and $< 0.1$ for perturbed radiances of $\pm 5\%$ and $\pm 10\%$, respectively, not dependent on SZA and wavelength. For higher aerosol load conditions (AOD $\sim 1.0$), we

have found nearly constant AOD differences of approximately -0.1 and -0.2 with SZA for negative radiance perturbations of -5% and -10% respectively. However, in the case of positive ZSR perturbations, we observed a SZA dependence, with higher AOD differences as SZA decreases, up to $\sim 0.35$ for a positive radiance perturbation of 10% and SZA of 20°.





### 4.1.2 RTM inputs sensitivity

- Surface Albedo sensitivity:

  The effect of the albedo uncertainty on the inferred AOD was also assessed for the same AOD and solar zenith angle range. We have performed the study for albedo values varying $\pm15\%$ of those values used in the generated lookup table. We have considered this variation range taking into account the results obtained by Tsvetsinskaya et al. (2006), who studied the spatial and temporal variability of surface albedo using MODIS data. They found very stable albedo values with a temporal variability essentially negligible in case of surfaces with high reflectivity, such as deserts, and spatial variations between 14% ($\lambda < 700$ mn) and 9% ($\lambda > 700$ nm). Consequently, a variability in surface albedo of 15% seems reasonable for our sensitivity study. The computed ZSR with the modified albedo values are used as inputs to retrieve the AOD and the result is compared with the actual AOD. The AOD differences for 440 and 870 nm, and several SZA and AOD values are shown in Table 2. This table shows that the surface reflectance effect on AOD is relatively low, with AOD differences somewhat higher with SZA, ranging from -0.016 to 0.017 for an AOD of 0.5, and from -0.037 to 0.040 for an AOD of 1.0. The wavelength dependence is also almost negligible.

- Mix of aerosol components sensitivity:

  In order to test the influence of the mixture of aerosol components in the inferred AOD, we have modified the assumed mixing ratio of the reference LUT for mineral dust and WASO components present in the boundary layer. Mineral dust components were perturbed in $\pm5\%$ of the slope coefficient in Eqs. 1, 2 and 3, and WASO components, assumed as a fixed value, were perturbed assuming a variation in concentration of $\pm50\%$. AOD differences considering perturbed and unperturbed values for the mix of aerosol components are presented in Fig. 3 for two AOD conditions (AOD $\sim$ 0.5 and 1.0) and two spectral ranges (440 and 870 nm). We have found negligible influence for AOD $\sim$ 0.5, with AOD differences ranging from $\pm0.02$ for all aerosol components, although it can be appreciated a slight dependence with SZA and wavelength. For higher aerosol content (AOD $\sim$ 1.0), we observed a little dependence with SZA and wavelength on AOD difference. We found AOD differences ranging from -0.1 to 0.05.

This sensitivity analysis showed that ZEN AOD combined standard uncertainty (determined by means of summation in quadrature of each term analyzed in the sensitivity study) is up to 0.06 (for AOD = 0.5) and 0.15 (for AOD = 1.0) when an instrumental error of 5% is considered. Higher AOD uncertainty is expected if instrumental error is >5%, up to 0.23 in the 10% limit of instrumental error (0.37 in case of very low SZAs).

## 5 RESULTS

### 5.1 ZEN-LUT method validation at three different AERONET sites

The performance of the ZEN-LUT methodology was tested in three different sites in which mineral dust plays an important role in their respective aerosol climatology (Izaña, Santa Cruz and Tamanrasset) using CE318 instruments as reference, since





it has been already widely tested under quite different conditions (Eck et al., 1999, 2001, 2003a, b, 2010; Holben et al., 2001, 2006; Dubovik et al., 2000). The CE318-LUT AOD was inferred after applying the ZEN-LUT methodology on ZSR data derived from PPL data provided by AERONET.

AOD data from Izaña and Santa Cruz stations were cloud screened using information from the SONA all-sky imager. Since any all-sky camera was available at Tamanrasset station, the cloud screening was performed by determining outliers in AOD using the modified Thompson Tau method (Cimbala , 2011).

We have restricted this analysis to desert dust events. We have identified the desert aerosol conditions following the criteria given in Guirado (2014), and Guirado et al. (2014) (Sect. 2.1). We have used AERONET data for the year 2013, as it was the most recent period of available AERONET 2.0 level AOD data in the three sites. The comparison analysis performed at the three stations at every available wavelength (440, 675 and 870 nm) is shown in Figs. 5, 6 and 7 and Table 3. We present the AOD CE318-LUT/CE318-AERONET scatterplot in Figs. 5 (a-c), 6 (a-c) and 7 (a-c), showing a good agreement for all channels in the three locations, with high correlations ($R^2 >0.95$). We have also found low RMSE values for Izaña and Santa Cruz (up to 0.011 and 0.021, respectively) but higher values for Tamanrasset (up to 0.035). The linear dependence between the two AOD datasets is different in the considered channels, with the lower slope the shorter wavelength at all sites, and more evident for Izaña. With regard to the AOD differences plotted in Figs. 5 (d-i), 6 (d-i) and 7 (d-i), it can be said that AOD is mostly underestimated by the ZEN-LUT method, with negligible dependence with SZA for the SZA range considered in this study. We have found maximum AOD differences up to -0.07 for Izaña and between -0.08 and 0.12 for Santa Cruz. These values are in agreement with the expected uncertainty involved in ZEN-LUT methodology AOD determination presented in Sect. 4.1. It also confirms the low instrumental errors affecting the CE318 (around ~5%). The considerably higher differences found for Tamarasset might be explained by the absence of a robust cloud screening method at this station.

## 5.2 ZEN-R41 and CE318 AERONET AOD comparison at Izaña

We have performed an intercomparison of AOD from ZEN-R41 and CE318 at Izaña station during year 2015. ZEN-R41 AOD, retrieved by applying the ZEN-LUT method to cloud screened ZSR data, was compared with AERONET level 1.5 AOD, for the 20°-65° SZA range. The results of the intercomparison for the four spectral bands are presented in Fig. 8 and Table 4. We present in Fig. 8 (a-d) a scatterplot between ZEN-R41 and CE318-AERONET AOD. A good correlation can be observed between both AODs, with $R^2$=0.97 and RMSEs ~ 0.026-0.027 for all channels, indicating that ZEN-R41 instrument and ZEN-LUT method altogether are adequate for AOD estimation. However, as we pointed out in previous subsection, the linear dependence between the two AOD datasets is different in the four channels. The AOD differences plotted in (Fig. 8e - 8l) shows a similar behavior to that presented in the former subsection for Izaña, showing that AOD is mostly underestimated, although in this case the differences found are larger as we have found maximum observed AOD differences up to 0.15 and RMSE values up to 0.030 (Table 4), but within the uncertainties estimated in section 4.1.



# 6 Conclusions

In this study we have presented the new ZEN system for dust AOD monitoring from ZSR observations. The ZEN system comprises the development of a new and robust radiometer (ZEN-R41) and a methodology to retrieve AOD from ZSR measurements through a lookup table (LUT) method (ZEN-LUT), especially designed for desert aerosols. This methodology, inspired by previous methodologies commonly applied to on-board satellite sensors, uses the radiative transfer code LibRad-tran and its packages to simulate ZSRs and the associated AODs. Then, AOD is inferred by minimizing a function which depends on the differences between simulated and measured ZSR at the corresponding solar zenith angle (SZA) in all the available wavelengths.

The main conclusions of this study are:

1. The comparison of ZSR from ZEN-R41 and CE318 showed a high coefficient of determination ($R^2$) for all wavelengths (0.99), although we observed relative mean bias (RMB) as high as -6.3% at 870 nm and 9.2% at 500 nm.

2. The sensitivity analysis, performed to identify the systematic errors (instrumental and radiative transfer inputs, or RTM) exerting the most influence on the final AOD, showed the instrumental errors and the aerosol model as the most important contributors to the final AOD uncertainty. This analysis estimated an AOD combined standard uncertainty in ZEN system up to 0.06 in case of AOD $\leq$ 0.5 and 0.15 for AOD up to 1.0, provided instrumental errors are minimized ($\sim$5%, as is the case of the CE318).

3. We have compared AOD from AERONET with AOD retrieved from CE318 ZSR by means of the ZEN-LUT method (CE318-LUT) in a common period in which AERONET level 2.0 is available at three stations (Izana, Santa Cruz and Tamanrasset). A good AOD agreement ($R^2$ from 0.99 at Santa Cruz to 0.95 at Tamanrasset) and RMSE values from 0.011 (at Izaña) to 0.035 (at Tamanrasset) have been obtained. The relatively worst results observed at Tamanrasset might be explained by the fact that no zenith-cloud screening was applied to the data of this station. We observed maximum AOD differences up to -0.07 at Izaña and between 0.08 and 0.12 for Santa Cruz, in agreement with the expected uncertainty involved in the ZEN-LUT methodology.

4. The AOD comparison at Izaña showed a good agreement between ZEN-R41 and AERONET ($R^2$ of 0.97), with observed AOD differences up to 0.15, and ZEN-R41 AOD systematically underestimated (mean bias ranging from -0.020 to -0.030). These results are also in agreement with the expected uncertainty for the ZEN system.

The results of this preliminary study indicate that the ZEN-LUT method is appropriate to infer dust AOD from ZSR measurements from the ZEN-R41, with an expected uncertainty between 0.06 and 0.15 in the AOD range between 0.5 and 1.0, which seems reasonable for most Saharan and Middle East sites affected by dust. However, a thorough validation with a higher number of ZEN-R41 radiometers installed in stations located in quite different environments affected by desert dust will be carried out in a near future to confirm and complement the results presented in this paper. The study of the impact caused by other aerosols in order to adapt this instrument to other environments, far from dust sources is an important issue to be ad-





dressed. Although clouds are not a major problem in desert regions for much of the year, autonomous cloud-screening system is being implemented into the ZEN-R41 radiometer in order to discriminate cloud contaminated ZSR data.

With the ZEN system we do not intend at all to replace accurate AOD measurements performed by sun-photometer networks (as AERONET) but to complement these observations in order to improve mineral dust monitoring in remote locations, where it

5    is difficult to deploy sun-photometers for logistical reasons and poor infrastructure. The ZEN system could be used individually with autonomous data processing, form networks with centralized data processing, or ultimately be incorporated into automatic weather stations in desert regions of an inexpensive and simple way. As a consequence, this instrument could play a key role in dust model data assimilation in near dust source regions, satellite validation and early warning within the WMO Sand and Dust Storm Warning Advisory and Assessment System (SDS WAS).

10    *Acknowledgements.* The authors are grateful to LibRadtran team for their assistance with the radiative transfer simulations performed in this paper. We also acknowledge to Izaña staff for maintaining the instrumentation thus ensuring the quality of data. This work has been developed within the framework of the activities of the World Meteorological Organization (WMO) Commission for Instruments and Methods of Observations (CIMO) Izaña Testbed for Aerosols and Water Vapor Remote Sensing Instruments. AERONET sun photometers at Izaña have been calibrated within the AERONET Europe TNA, supported by the European Community-Research Infrastructure Action under the FP7

15    ACTRIS grant agreement no. 262254.



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





**Table 1.** Coefficient of determination ($R^2$), relative root mean square error (RRMSE), in %, relative mean bias (RMB), in %, and number of coincident data (N) for the ZSR comparisons between CE318 and ZEN-R41 measurements at four different spectral bands (440, 500, 675 and 870 nm) performed at Izaña in 2015.

| Wavelength | $R^2$ | $RRMSE$ | $RMB$ | $N$ |
|---|---|---|---|---|
| 870 | 0.99 | 4.7 | -6.3 | 616 |
| 675 | 0.99 | 7.2 | 6.7 | 616 |
| 500 | 0.99 | 4.3 | 9.2 | 616 |
| 440 | 0.99 | 3.4 | 5.1 | 616 |

**Table 2.** Sensitivity study for an albedo perturbation of ±15%. AOD differences between perturbed and unperturbed situations ($\Delta AOD$) for 440 and 870 nm spectral bands are shown for two AOD values (0.5 and 1.0) and different solar zenith angles (SZA) ranging from 20° to 65°.

| $AOD$ | $SZA$ | $\Delta AOD_{440}$ | $\Delta AOD_{870}$ |
|---|---|---|---|
| 0.5 | 20 | -0.006/0.006 | -0.006/0.007 |
|  | 30 | -0.010/0.010 | -0.011/0.011 |
|  | 40 | -0.013/0.013 | -0.013/0.013 |
|  | 50 | -0.015/0.016 | -0.015/0.017 |
|  | 60 | -0.017/0.017 | -0.017/0.017 |
|  | 65 | -0.016/0.017 | -0.016/0.017 |
| 1.0 | 20 | -0.016/0.017 | -0.016/0.018 |
|  | 30 | -0.023/0.024 | -0.025/0.026 |
|  | 40 | -0.028/0.030 | -0.031/0.033 |
|  | 50 | -0.032/0.033 | -0.035/0.038 |
|  | 60 | -0.033/0.036 | -0.037/0.040 |
|  | 65 | -0.033/0.034 | -0.037/0.040 |





**Table 3.** Coefficient of determination ($R^2$), root mean square error (RMSE), mean bias (MB), and number of coincident data (N) for the AOD comparison between CE318-AERONET and CE318-LUT at three different spectral bands (440, 675 and 870 nm) performed at Izaña (IZO), Santa Cruz (SCO) and Tamanrasset (TAM) stations in 2013.

| Station | Wavelength (nm) | $R^2$ | $RMSE$ | $MB$ | $N$ |
|---------|-----------------|-------|--------|------|-----|
| IZO | 870 | 0.97 | 0.011 | -0.018 | 180 |
|  | 675 | 0.98 | 0.010 | -0.020 | 180 |
|  | 440 | 0.97 | 0.011 | -0.018 | 180 |
| SCO | 870 | 0.99 | 0.021 | -0.021 | 210 |
|  | 675 | 0.99 | 0.021 | -0.019 | 210 |
|  | 440 | 0.99 | 0.021 | -0.020 | 210 |
| TAM | 870 | 0.95 | 0.034 | -0.023 | 385 |
|  | 675 | 0.95 | 0.033 | -0.012 | 385 |
|  | 440 | 0.94 | 0.035 | -0.009 | 385 |

**Table 4.** Coefficient of determination ($R^2$), root mean square error (RMSE), mean bias (MB), and number of coincident data (N) for the AOD comparisons between CE318-AERONET and ZEN-R41 at four different spectral bands (440, 500, 675 and 870 nm) performed at Izaña station in 2015.

| Wavelength (nm) | $R^2$ | $RMSE$ | $MB$ | $N$ |
|-----------------|-------|--------|------|-----|
| 870 | 0.97 | 0.026 | - 0.020 | 616 |
| 675 | 0.97 | 0.026 | -0.025 | 616 |
| 500 | 0.97 | 0.026 | -0.029 | 616 |
| 440 | 0.97 | 0.027 | -0.030 | 616 |



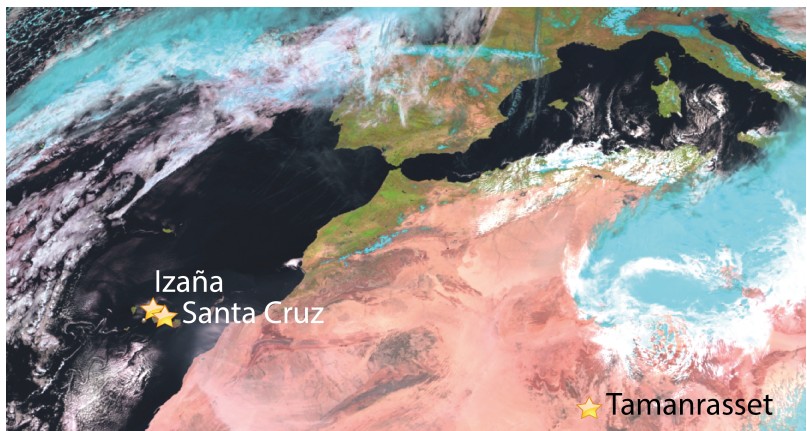

**Figure 1.** Meteosat/TERRA image showing a Saharan dust outbreak over the study area on 12 January 2015, where the Izaña, Santa Cruz and Tamanrasset sites are indicated with yellow stars.

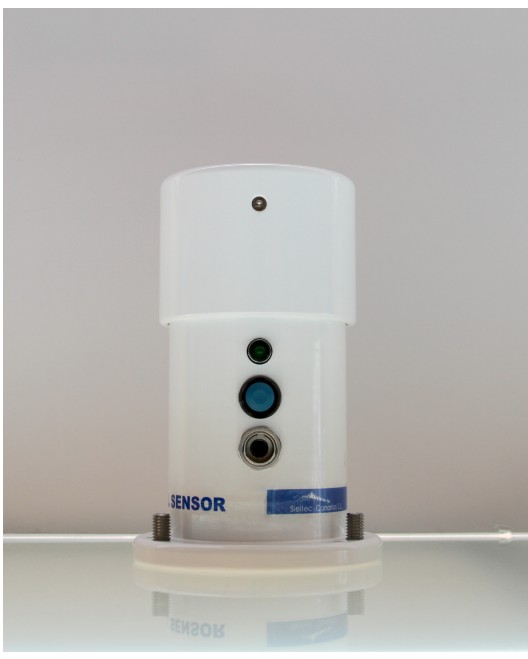

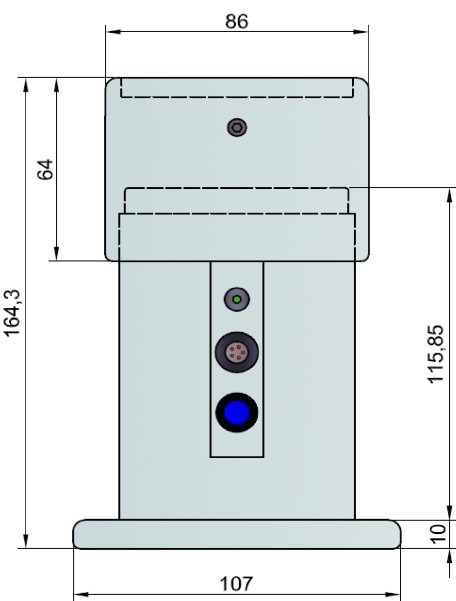

**Figure 2.** ZEN-R41 scheme and dimensions.





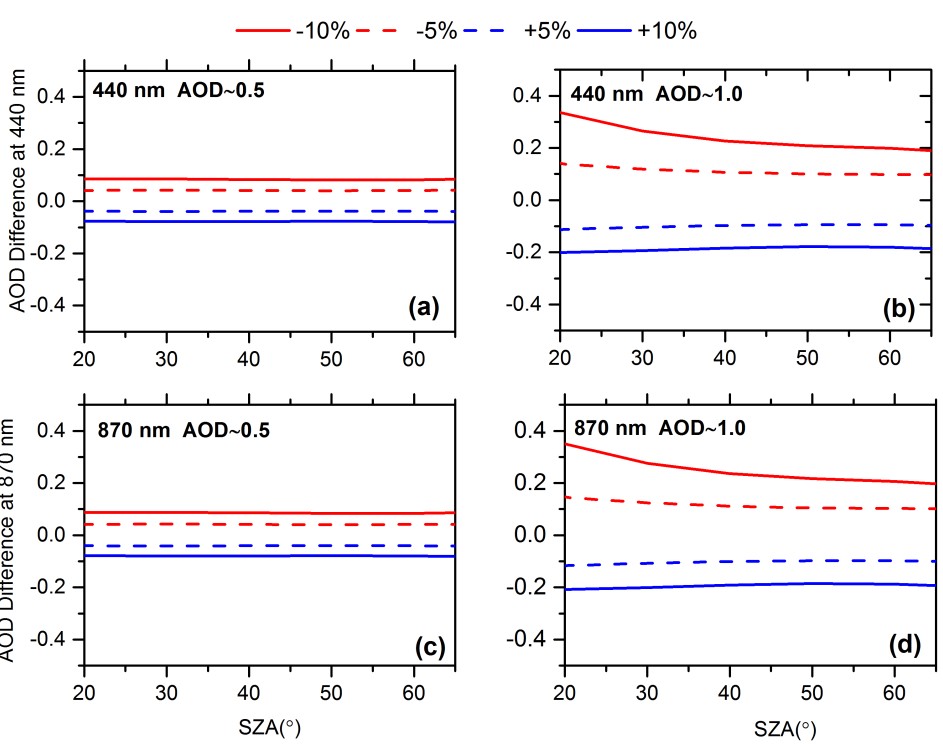

**Figure 3.** AOD difference versus SZA for different AOD conditions (0.5 and 1.0) and ZSR perturbed ±5% and ±10%, for 440 nm (**a**, **b**) and 870 nm (**c**, **d**).





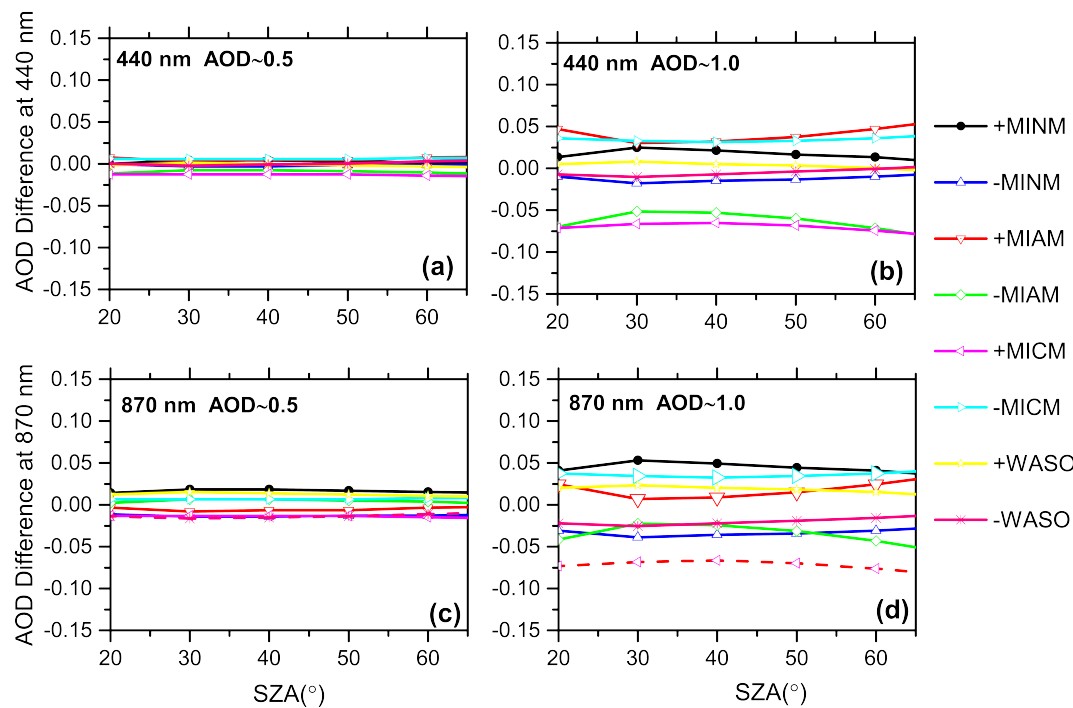

**Figure 4.** AOD difference versus SZA for different AOD (0.5 and 1.0) and 440 nm (**a**, **b**) and 870 nm (**c**, **d**) wavelengths. We have included different concentrations of four different aerosol components, MINM, MIAM and MICM perturbed in ±5% and WASO concentration perturbed in ±50%.





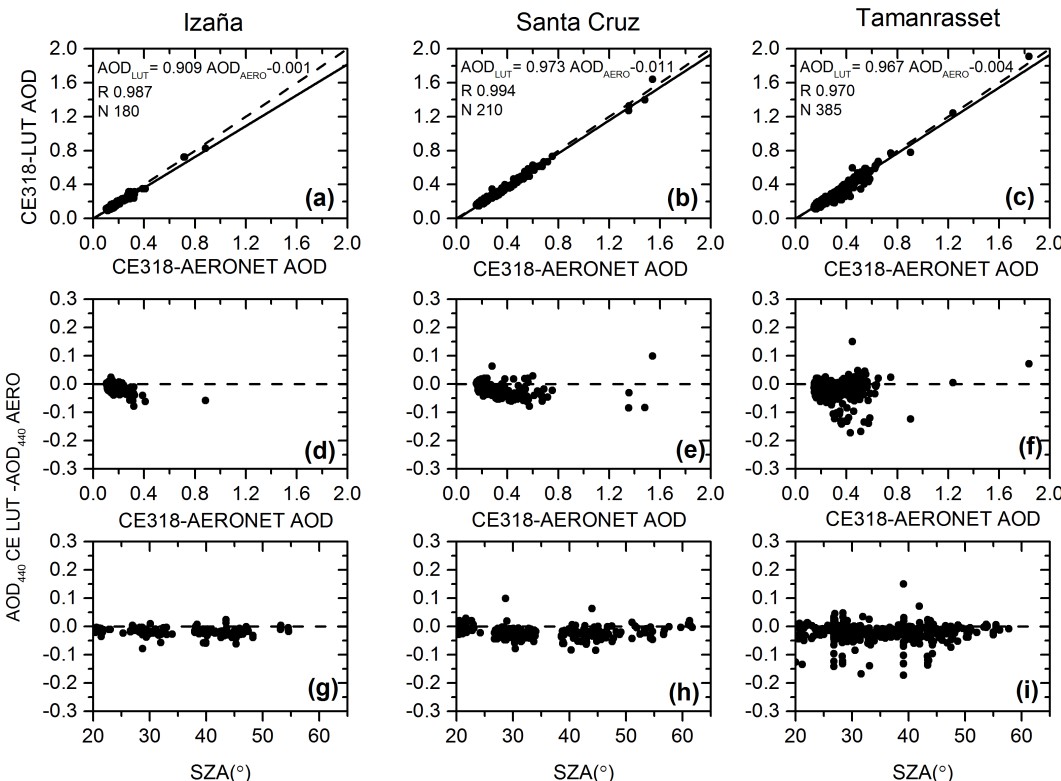

**Figure 5.** AOD scatterplot at 440 nm between CE318-AERONET and CE318-LUT for Izaña (**a**), Santa Cruz (**b**) and Tamanrasset (**c**) stations in 2013 for $20° < SZA < 65°$ (**a, b, c**). The black solid lines are the least-square fits, and the dashed lines are the diagonals ($y = x$). The least-square fit parameters are shown in the legend (slope, intercept, correlation coefficient (R) and number of data (N)). AOD difference in 440 nm between CE318-AERONET and CE318-LUT respect to CE318-AERONET (**d, e, f**) and SZA (°) (**g, h, i**), respectively.





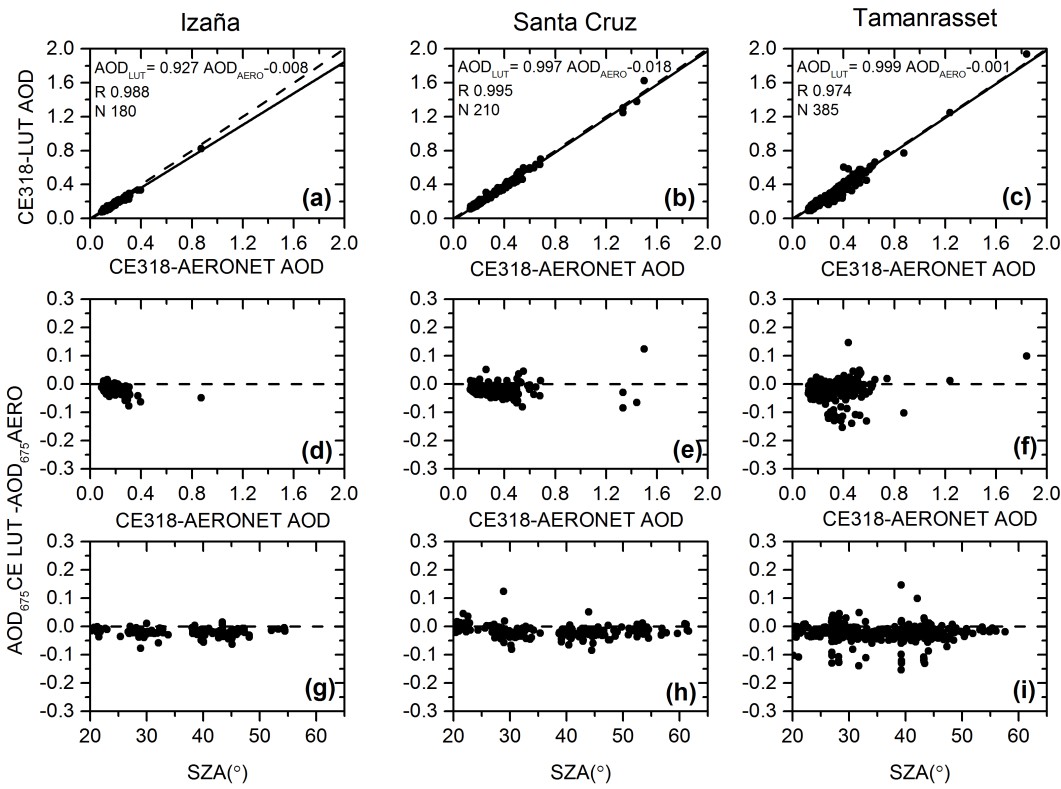

**Figure 6.** AOD scatterplot at 675 nm between CE318-AERONET and CE318-LUT for Izaña (**a**), Santa Cruz (**b**) and Tamanrasset (**c**) stations in 2013 for $20° < SZA < 65°$ (**a**, **b**, **c**). The black solid lines are the least-square fits, and the dashed lines are the diagonals ($y = x$). The least-square fit parameters are shown in the legend (slope, intercept, correlation coefficient (R) and number of data (N)). AOD difference in 675 nm between CE318-AERONET and CE318-LUT respect to CE318-AERONET (**d**, **e**, **f**) and SZA (°) (**g**, **h**, **i**), respectively.





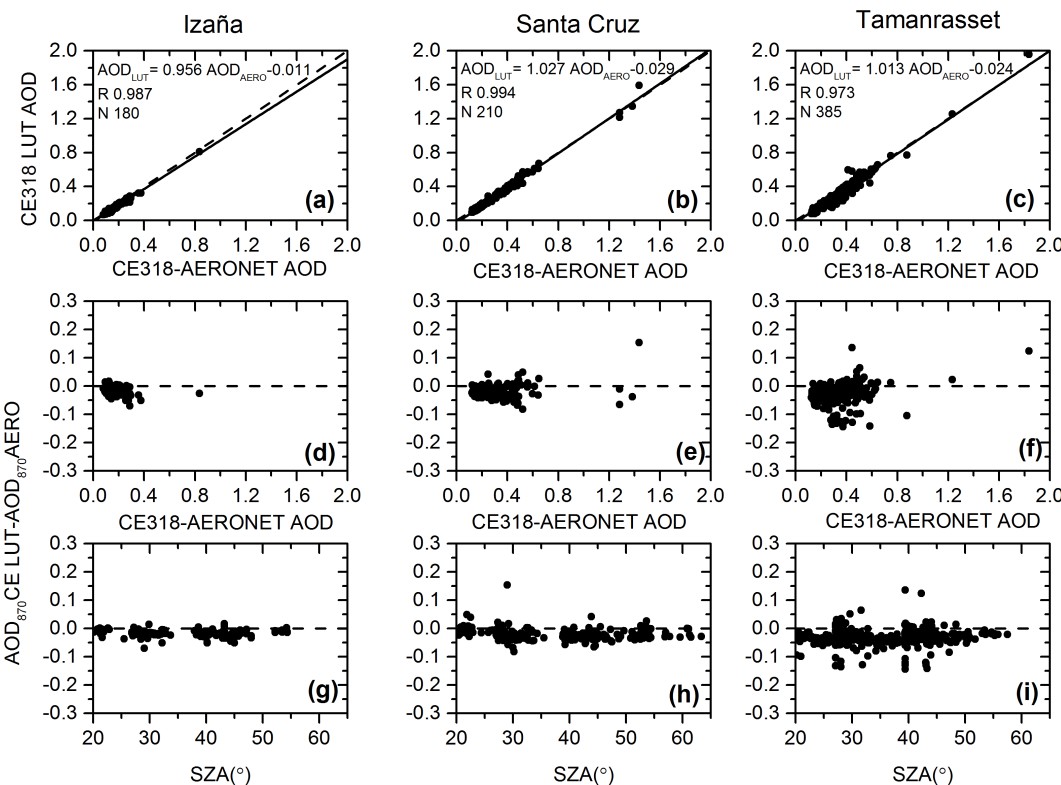

**Figure 7.** AOD scatterplot at 870 nm between CE318-AERONET and CE318-LUT for Izaña (**a**), Santa Cruz (**b**) and Tamanrasset (**c**) stations in 2013 for 20°<SZA<65° (**a, b, c**). The black solid lines are the least-square fits, and the dashed lines are the diagonals ($y = x$). The least-square fit parameters are shown in the legend (slope, intercept, correlation coefficient (R) and number of data (N)). AOD difference in 870 nm between CE318-AERONET and CE318-LUT respect to CE318-AERONET (**d, e, f**) and SZA (°) (**g, h, i**), respectively.





**Figure 8.** AOD comparisons between CE318-AERONET and ZEN-R41 for four different spectral bands (440, 500, 675 and 870 nm) performed at Izaña station in 2015. In the upper panel (**a–d**) AOD scatterplots AERONET/ZEN41 are presented. The middle panel (**e–h**) shows the AOD differences versus AERONET AOD. AOD differences versus solar zenith angle (SZA in $^{\circ}$) are shown in the lower panel (**i–l**).