# Peer review of "A new zenith looking narrow-band radiometer based system (ZEN) for dust Aerosol Optical Depth monitoring"

_Atmospheric Measurement Techniques, 2016_

## Referee Comment (RC1) · Anonymous Referee #1 · 10 Oct 2016

The author presented a new zenith looking narrow-band radiometer based system (ZEN) to monitor the dust aerosol optical depth with a new radiometer (ZEN-R41) and a methodology for AOD retrieval (ZEN-LUT). A comparison experiment was carried out with the AERONET data at three sites characterized by a regular presence of desert mineral dust aerosols: Izana and Santa Cruz in the Canary Islands, and Tamanrasset in Algeria and the results were very well. This is a very nice work because it offers a new method to measure AOD. I suggest this paper published in AMT just with some questions as following:

Q1. Page 1-2. In the introduction of aerosol properties,It would be better to cite some new references such as related studies e.g. SKYNET, CARSNET, WMO-PFR, etc. Q2.

Page 3 line 34. "...with AOD values < 0.15." Which wavelength of AOD here? Q3. Page 9 line 28. "In case of AOD ~0.5...". which wavelength of AOD you used in this study? Q4. Page 11 line 15. "...it can be said that AOD is mostly underestimated by the ZEN-LUT method." What's the probable reason? Q5. Figure 3. The top major ticks of Fig (b) are different from other pictures, please revise it.

---

## Referee Comment (RC2) · Anonymous Referee #2 · 26 Oct 2016

General comments:

The manuscript presents a new aerosol measurement system consisting on a zenith looking narrow-band radiometer (ZEN-R41) and an AOD retrieval methodology based on look-up tables (ZEN-LUT) computed with the LibRadTran model and OPAC database.

The study includes a sensitivity analysis of the retrieved AOD with instrumental and RTM input uncertainties, and a validation against state of the art sunphotometers such as Cimel CE318 in different sites dominated by dust conditions.

The instrument does not have mobile parts, being robust and therefore, well suited for

the applications named in the manuscript. The results for the three sites are relatively good, taking into account the system limitations.

The text, figure and tables are concise but clear. The English style is readable, without major grammar or ortographic flaws.

This manuscript is also very suited for AMT journal, so my recommendation is to accept the paper after some suggestions are addressed.

Specific questions and comments:

- In the abstract, a mention to the sensitivity analysis with an estimation of the inherent system uncertainty would be useful.

- Page 5, line 4: how do you interpolate the radiance in the PPL for retrieving the ZSR? Do you use only two or more angles? Linear or non-linear approach?

- Page 5, line 12: please add some details about the cloud cover algorithm, if they are not included in the previous reference

- Dusting of the sensor windows could be an error source, mainly when the window is horizontal and unprotected under strong dust episodes. How this issue has been addressed in the three sites?

- Page 6, line 13: The calibration of the instrument, has been performed for only one radiance level?

- Page 6, line 20: how the ZEN FOV has been taken into account when building up the LUT? This could be more important when the SZA is lower, as the variation of the radiation field within the instrument FOV is larger.

- Page 11, line 6: in tamanrasset site it is not possible to screen the clouds in the zenith because there is no sky camera available. Therefore, the authors apply the TT method to the AOD series. This is a valid solution. However, later it is said that the higher differences found in Tamanrasset could be due to the abscence of a robust
cloud screening method. Why not using the PPL information to improve the results for Tamanrasset data? I suggest using the information derived from the interpolation of radiance at the zenith (that's one more reason to use more than two points for the interpolation, as suggested before) as an auxiliary tool for rejecting ZSR data affected by clouds. Otherwise, could deposited dust be another reason for a higher deviation at this site?

Technical or minor comments:

- Page 2, lines 35: although the reviewer agrees that the tracking system from cimel increases the power consumption, solar panels are sufficient to feed the instrument, without a need to be connected to an electric grid.

- Page 3, lines 27-30. It looks like a very long sentence, consider to rewrite it in two different sentences.

- page 4, line 24: consider more recent references for the Dubovik code.

- page 4, line 29: to be accurate, the measurements are performed in the solar almucantar plane, where the almucantar zenith angle is equal to the solar zenith angle. Same for PPL.

- Page 6, line 27: but still within the combined uncertainty from ZEN and Cimel
- Page 10, line 11: this information could be shown in a Figure (as with the other cases)
- Page 10, line 19: Figure 3 -> 4?
- page 11, line 20: TamaNrasset
- Sometimes R2 is used in the text, R in the figures. Please use the same.

---

## Author Comment (AC1) · 28 Dec 2016

**Response to Referee #1**

We thank to Referee #1 for his/her positive and constructive comments. The responses to every question and comment are shown below.

***Q1. Page 1-2. In the introduction of aerosol properties, it would be better to cite some new references such as related studies e.g. SKYNET, CARSNET, WMO-PFR, etc.***

The authors agree with the referee comment. These networks have been referenced in page 2 line 26.

***Q2. Page 3 line 34. "…with AOD values < 0.15." Which wavelength of AOD here?***

We are referring to 675 nm. We have included the wavelengths.

***Q3. Page 9 line 28. "In case of AOD≈0.5 …". which wavelength of AOD you used in this study?***

We are referring to 500 nm. Included

***Q4. Page 11 line 15. "…it can be said that AOD is mostly underestimated by the ZEN-LUT method." What's the probable reason?***

As we pointed out in the sensitivity study, the cause of the observed ZSR differences might come from the instrumental side or from the RTM inputs. We discard stray light as a possible explanation as the CE318 collimator performs a $10^{-5}$ stray light rejection for scattering angles=$3^{o}$ (Holben et al. 1998). Other instrument-related errors, as dirtiness over lenses or a calibration offset normally produce a systematic ZSR overestimation or underestimation at all scattering angles what does not agree with the observed positive ZSR differences for SZA below 20° and negative differences for SZA larger than 20°. For the same reason, we also discard a wrong input albedo as a possible cause.

We suspect that the main cause of the systematic AOD underestimation might be found in the aerosol optical properties derived from the aerosol model chosen to produce the look-up tables of Zenith Sky Radiances (ZSRs). To illustrate this hypothesis we have performed a comparative analysis of the measured ZSR with several simulated ZSR performed for the sample days, using the actual lookup table and the 4 individual OPAC mineral components (MINM, MITR, MIAM and MICM) with phase functions and single scattering albedos (SSA) taken from the LibRadtran spheroids components (see table 1 for the SSA values). In figure 2, the relative differences between measured and computed ZSR for the different cases, are represented against SZA.  It can be seen larger differences for the extreme cases, i.e. supposing we only have fine mineral particles (MINM) or large mineral particles (MICM), but smaller when intermediate size mineral particles are employed (MITR and MIAM).  The lookup table for Izaña is composed by a mix of WASO (fine particles), MINM (fine particles) and MITR (intermediate size particles) and it shows slightly larger negative differences (especially for 675 and 870 nm wavelengths) than considering just the presence of  MITR or MIAM components. It might indicate that chosen mix of components to generate the lookup table has an excess of fine particles.

[Figure]

Figure 1. Difference between measured and computed ZSR (lookup table) at three wavelengths (440(a), 675(b) and 870(c)) for two days (23[rd] and 24[th] of August 2013) with similar and stable AOD conditions (<$AOD_{440}$>≈0.3).

[Figure]

Figure 2. Relative difference between measured and computed ZSR at three wavelengths (440(a), 675(b) and 870(c)) for two days (23[rd] and 24[th] of August 2013) with similar and stable AOD conditions (<$AOD_{440}$>≈0.3). Every different mark represents the relative difference using different optical properties to compute the ZSR.

| | SSA(450nm) | SSA(650nm) | SSA(900nm) |
|---|---|---|---|
| MINM | 0.95 | 0.97 | 0.98 |
| MITR | 0.84 | 0.92 | 0.95 |
| MIAM | 0.81 | 0.91 | 0.94 |
| MICM | 0.61 | 0.71 | 0.76 |

Table 1. Single scattering albedo values for four mineral aerosol OPAC components (spheroids) taken from LibRadtran at three different wavelengths.

*Q5. Figure 3. The top major ticks of Fig (b) are different from other pictures, please revise it.*

Done.

**REFERENCES**

Holben, B. N., Eck, T. F., Slutsker, I., Tanré, D., Buis, J. P., Setzer, A., Vermote, E., Reagan, J. A., Kaufman, Y. J., Nakajima, T., Lavenu, F., Jankowiak, I. and Smirnov, A.: AERONET- A federated instrument network and data archive for aerosol characterization, Rem. Sens. Environ., 66, 1-16, 1998.

---

## Author Comment (AC2)

**Response to Referee #2**

We thank to Referee #2 for his/her positive review and helpful comments. Following, are the authors answer.

**Specific questions and comments:**

*- In the abstract, a mention to the sensitivity analysis with an estimation of the inherent system uncertainty would be useful.*

We have included the following sentence in the abstract: "The sensitivity study proved that the ZEN-LUT method is appropriate to infer AOD from ZSR measurements with an AOD standard uncertainty up to 0.06 for $AOD_{500nm} \approx 0.5$, and up to 0.15 for $AOD_{500nm} \approx 1.0$, considering instrumental errors of 5%.".

*- Page 5, line 4: how do you interpolate the radiance in the PPL for retrieving the ZSR? Do you use only two or more angles? Linear or non-linear approach?*

The ZSR has been retrieved by linear interpolation between two adjacent points. A cubic spline interpolation has been also tested, obtaining very similar results.

*- Page 5, line 12: please add some details about the cloud cover algorithm, if they are not included in the previous reference.*

Since the cloud detection was performed visually we have rewritten the paragraph as follows: "Cloud cover detection was performed by visual inspection analyzing each individual hemispheric image…"

*- Dusting of the sensor windows could be an error source, mainly when the window is horizontal and unprotected under strong dust episodes. How this issue has been addressed in the three sites?*

In this study, only one ZEN-R41 prototype has been evaluated at Izaña station, where strict maintenance protocol is carried out by the observers, with frequent diurnal checks, so the cleanness of the window is assured. For those places with limited access, we propose an additional external blower to remove dust from the external window, as mentioned in page 5 line 27.

We have focused on the ZEN-R41 performance, ensuring a maximum cleaning, in order to assess the feasibility of the instrument and the methodology used. Operationally, it is true that it will be necessary to have the external blower and assess its effectiveness. This will be addressed in a future paper in which an assessment of the measurements obtained with a ZEN-R41 at Tamanrasset will be shown.

*- Page 6, line 13: The calibration of the instrument, has been performed for only one radiance level?*

Yes, the integrating sphere employed in this work is calibrated only for one radiance level so the radiometer cannot be calibrated for other radiance levels. However, the results of the

radiometric comparison between CE318 and ZEN-R41 shown in section 3.1, indicate that both instruments perform quite similar ZSR measurements (within the uncertainty of the calibration technique) with a high coefficient of determination ($R^2$=0.99) and therefore a non-linearity problem of the ZEN-R41 detectors can be discarded. As a result we expect a very similar ZEN-R41 response if it was calibrated using a sphere calibrated at other radiance levels.

***- Page 6, line 20: how the ZEN FOV has been taken into account when building up the LUT? This could be more important when the SZA is lower, as the variation of the radiation field within the instrument FOV is larger.***

In order to simplify calculations, we have not taken the instrument FOV into account. Previous works, Shiobara et al., 1991 and B. Torres PhD theses, 2012, demonstrated that the consideration of a finite FOV (6$^o$ or lower) in the radiance measurement is almost negligible for scattering angles larger than 15$^o$, being considerable for small scattering angles, as the referee already indicated. Shiobara et al. 1991, performed a test of the influence of different FOV values (from 0.25$^o$ to 6$^o$) on the phase function. They found relative errors below 1% in the phase function for scattering angles larger than 10$^o$ and a FOV of 6$^o$, while for scattering angles lower than 2$^o$ and a 6$^o$ FOV, the relative error are higher than -10%. A similar analysis can found in B. Torres 2012 PhD theses, where the influence of two finite FOV sizes, 1.2$^o$ and 2.4$^o$, on the measured radiance at several spectral bands (centered at 440, 670, 870 and 1020 nm) were tested. This author found negligible influence for scattering angles larger than 15$^o$ and a 2.4$^o$ FOV size, while larger than 2% for scattering angles lower than 3$^o$. Considering these two previous works, we can say that the influence of the ZEN-R41 finite FOV (≈3$^o$) on the measured ZSR for SZA larger than 20$^o$, which is the range used in this work, is negligible.

***- Page 11, line 6: in Tamanrasset site it is not possible to screen the clouds in the zenith because there is no sky camera available. Therefore, the authors apply the TT method to the AOD series. This is a valid solution. However, later it is said that the higher differences found in Tamanrasset could be due to the absence of a robust cloud screening method. Why not using the PPL information to improve the results for Tamanrasset data? I suggest using the information derived from the interpolation of radiance at the zenith (that's one more reason to use more than two points for the interpolation, as suggested before) as an auxiliary tool for rejecting ZSR data affected by clouds. Otherwise, could deposited dust be another reason for a higher deviation at this site?***

The authors agree with the referee that the PPL information can be used to check cloud contaminated data. However, for the authors it is not evident using the interpolation to perform cloud screening. On the other hand, we propose to check the smoothness of the PPL curves to detect clouds as commented in Holben et al. (1998). To do this, we have checked for the second derivative of the PPL Radiances with respect to the scattering angle (we do not use all scattering angles available in the PPL scenario, just from 2$^o$ to 90$^o$). For clear skies the second derivative of the PPL curve will be positive for the considered scattering angles, but when clouds are present this value might become negative for some scattering angles. Then, if negative values are found, at least for one scattering angle, the PPL curve is not used to obtain the ZSR. The threshold value for this smoothness criterion is not exactly 0 but -1E-7, as determined empirically for the 3 spectral bands.

In table 1, the statistics results for Tamanrasset station after applying the PPL smoothness and the Thompson Tau cloud screening methods, are shown. As it can be seen from the statistics, the PPL Smoothness criterion seems to be more restrictive than Thompson Tau method, since 321 and 385 data points are selected as cloud-free, respectively, and the comparisons with AERONET AOD data are improved, showing higher $R^2$ and lower RMSE. So, it can be concluded that the PPL smoothness criterion shows a better cloud contaminated data rejection. However, further independent evaluations are needed.

The authors concur with the referee that deposited dust on the CE318's lenses might be responsible of part of the AOD differences observed in the comparisons between the ZEN-LUT method and AERONET. To illustrate such statement, the AOD results (cloud screened) and the AOD differences between both methods for a period of time of 12 days in June (from 9th to 20th of June), when AOD absolute differences higher than 0.1 were found, are shown in figure 1. In this figure it can be seen a good agreement between both methodologies for the period between 9th and 12th of June, with low absolute AOD differences, but higher discrepancies after 13th of June, when an abrupt change can be observed. It can be also seen a diurnal cycle in AERONET AOD data for the period of time between 13th and 18th of June. The authors suspect that the observed concave shape of the diurnal AOD shown by AERONET in that period of time is fictitious and it is produced by dirt on the lenses. Dirty lenses produce a reduction on the measured direct sun light, leading to a magnification in the retrieved AOD which increases as the SZA decreases. On the other side, the dirtiness over lenses causes also a decrease of the measured sky radiance which results in a reduction of the retrieved AOD by ZEN-LUT method. Then, as the effect produced by dirty lenses on the retrieved AOD is opposite in both methods, the AOD differences are higher than they should be with clean lenses.

The cloud detection algorithm has been changed in the final version taking into account this comment. We have used the PPL Smoothness criterion to screen clouds in Tamanrasset.

| Method | Wavelength (nm) | $R^2$ | RMSE | MB | N |
|---|---|---|---|---|---|
| PPL Smoothness | 870 | 0.98 | 0.030 | -0.031 | 321 |
| | 675 | 0.98 | 0.030 | -0.022 | 321 |
| | 440 | 0.97 | 0.032 | -0.023 | 321 |
| Thompson Tau | 870 | 0.95 | 0.034 | -0.023 | 385 |
| | 675 | 0.95 | 0.033 | -0.012 | 385 |
| | 440 | 0.94 | 0.035 | -0.009 | 385 |

Table 1. Statistic results for Tamanrasset using two different cloud screening methods: PPL smoothness and the modified Thompson Tau method.

[Figure]

Figure 1. (a)Temporal evolution of the AOD at 675 nm for a period of time included between 2013/06/09 and 2013/06/21 (AERONET in red dots and ZEN-LUT in blue circles). (b) Temporal evolution of the AOD difference at 675 nm between AERONET and the ZEN-LUT technique for the same period of time.

**Technical or minor comments:**

*- Page 2, lines 35: although the reviewer agrees that the tracking system from cimel increases the power consumption, solar panels are sufficient to feed the instrument, without a need to be connected to an electric grid.*

The corresponding sentence has been removed since this is not an essential limitation.

*- Page 3, lines 27-30. It looks like a very long sentence, consider to rewrite it in two different sentences.*

We have rewritten the sentence as follows: "The Izaña… background conditions. It is a consequence of its location over a strong temperature inversion layer as a result of general subsidence processes and the presence of cool trade winds in lower levels."

*- page 4, line 24: consider more recent references for the Dubovik code.*

We have included the following references:

Dubovik, O., Holben, B., Lapyonok, T., Sinyuk, A., Mishchenko, M., Yang, P., and Slutsker, I.: Non-spherical aerosol retrieval method employing light scattering by spheroids, Geophysical Research Letter, 29, doi:{10.1029/2001GL014506}, 2002b.

Dubovik, O., Sinyuk, A., Lapyonok, T., Holben, B. N., Mishchenko, M., Yang, P., Eck, T., Volten, H., Munoz, O., Veihelmann, B., Van Der Zande, W. J., Leon, J., Sorokin, M., and Slutsker, I.: Application of spheroid models to account for aerosol particle nonsphericity in remote sensing of desert dust, Journal of Geophysical Research Atmospheres, 111, doi:{10.1029/2005JD006619}, 2006.

*- page 4, line 29: to be accurate, the measurements are performed in the solar almucantar plane, where the almucantar zenith angle is equal to the solar zenith angle.*

*Same for PPL.*

We consider this information is included in the text. In Page 4, line 30 we wrote: "In the ALM routine, the azimuth angle is varied while the zenith angle is kept constant (equals to the solar zenith angle)". Maybe we have not understood the question...

Same for PPL.

*- Page 6, line 27: but still within the combined uncertainty from ZEN and Cimel*

We have added this information.

*- Page 10, line 11: this information could be shown in a Figure (as with the other cases)*

We prefer to present the results for albedo effect in a table, considering the amount of figures included in this manuscript.

*- Page 10, line 19: Figure 3 -> 4?*

It was a typo. We are referring to Fig. 4.

*- page 11, line 20: TamaNrasset*

 It is shown in figure 1. Corrected.

*- Sometimes R2 is used in the text, R in the figures. Please use the same*

We have replaced R by R2 in the figures 5, 6, 7 and 8.

**REFERENCES**

Holben, B. N., Eck, T. F., Slutsker, I., Tanré, D., Buis, J. P., Setzer, A., Vermote, E., Reagan, J. A., Kaufman, Y. J., Nakajima, T., Lavenu, F., Jankowiak, I. and Smirnov, A.: AERONET- A federated instrument network and data archive for aerosol characterization, Rem. Sens. Environ., 66, 1-16, 1998.

Shiobara, M., Tadahiro Hayasaka, T. Nakajima and M. Tanaka, 1991. Aerosol monitoring using a scanning spectral radiometer in Sendai, Japan. J. Meteorol. Soc. of Japan, 60: 57-70.

Torres, B.: Study on the influence of different error sources on sky radiance measurements and inversion-derived aerosol products in the frame of AERONET, Ph.D. thesis, Universidad de Valladolid, 2012.

---

## Author Comment (AC3)

[revised manuscript text omitted]
 but wrong data can also be detected by checking the smoothness of the PPL curve (Holben et al., 1998). In the present study, the CE318 ZSR values are obtained by linear interpolation of the PPL data to the zenith position (it is not possible to do it with ALM measurements). To detect and remove the presence of clouds in data, we have visually checked hemispheric images in those places where an all sky camera was available. In case of an all sky camera was not available at the station, a smoothness criterion was applied on the PPL curve to detect clouds. This smoothness criterion is based on the analysis of the second derivative of the PPL randiances with respect the scattering angle. The data is considered as cloud contaminated if the second derivative is negative at any scattering angle between $2°$ and $90°$. The threshold value for this smoothness criterion is not 0 but $-1 \cdot 10^{-7}$, as determined empirically.

[revised manuscript text omitted]

Wehrli, C., 2005. GAWPFR: A network of Aerosol Optical Depth observations with Precision Filter Radiometers. In: WMO/GAW Experts workshop on a global surface based network for long term observations of column aerosol optical properties, GAW Report No. 162, WMO TD No. 1287.

Winker, D., Pelon, J., Coakley, J., Ackerman, S., Charlson, R., Colarco, P., Flamant, P., Fu, Q., Hoff, R., Kittaka, C., Kubar, T., Treut, H. L., McCormick, M., Megie, G., Poole, L., Powell, K., Trepte, C., Vaughan, M., and Wielicki, B.: The CALIPSO Mission: a global 3D view of aerosols and clouds, B. Am. Meteor. Soc., 91, 1211-1229, 2010.

World Meteorological Organization, Report of the WMO workshop on the measurement of atmospheric optical depth and turbidity. Silver Spring, Maryland, 6-10 December 1993, Bruce Hicks (Ed.), obtainable from WMO as Technical Document No. 659 (Geneva, Switzerland), www.wmo.ch, 28 pp, 1993.

World Meteorological Organization: Commission for Instruments and Methods of Observation, Sixteenth session WMO no.1138, Saint Petersburg, Secretariat of the World Meteorological Organization, 2014.

Zhang, J. and Reid, J. S.: MODIS Aerosol Product Analysis for Data Assimilation: Assessment of Level 2 Aerosol Optical Thickness Retrievals, J. Geophys. Res, 111, D22207, doi:10.1029/2005JD006898, 2006.

Zhang, J. and Reid, J.: A decadal regional and global trend analysis of the aerosol optical depth using a data-assimilation grade over-water MODIS and level 2 MISR aerosol products, Atmos. Chem. Phys, 10, 10949-10963, 2010.

**Table 1.** Coefficient of determination ($R^2$), relative root mean square error (RRMSE), in %, relative mean bias (RMB), in %, and number of coincident data (N) for the ZSR comparisons between CE318 and ZEN-R41 measurements at four different spectral bands (440, 500, 675 and 870 nm) performed at Izaña in 2015.

| Wavelength | $R^2$ | $RRMSE$ | $RMB$ | $N$ |
|:---:|:---:|:---:|:---:|:---:|
| 870 | 0.99 | 4.7 | -6.3 | 616 |
| 675 | 0.99 | 7.2 | 6.7 | 616 |
| 500 | 0.99 | 4.3 | 9.2 | 616 |
| 440 | 0.99 | 3.4 | 5.1 | 616 |

**Table 2.** Sensitivity study for an albedo perturbation of $\pm15\%$. AOD differences between perturbed and unperturbed situations ($\Delta AOD$) for 440 and 870 nm spectral bands are shown for two AOD values ( $AOD_{500nm} \sim 0.5$ and 1.0) and different solar zenith angles (SZA) ranging from $20°$ to $65°$.

| $AOD$ | $SZA$ | $\Delta AOD_{440}$ | $\Delta AOD_{870}$ |
|:---:|:---:|:---:|:---:|
| 0.5 | 20 | -0.006/0.006 | -0.006/0.007 |
|  | 30 | -0.010/0.010 | -0.011/0.011 |
|  | 40 | -0.013/0.013 | -0.013/0.013 |
|  | 50 | -0.015/0.016 | -0.015/0.017 |
|  | 60 | -0.017/0.017 | -0.017/0.017 |
|  | 65 | -0.016/0.017 | -0.016/0.017 |
| 1.0 | 20 | -0.016/0.017 | -0.016/0.018 |
|  | 30 | -0.023/0.024 | -0.025/0.026 |
|  | 40 | -0.028/0.030 | -0.031/0.033 |
|  | 50 | -0.032/0.033 | -0.035/0.038 |
|  | 60 | -0.033/0.036 | -0.037/0.040 |
|  | 65 | -0.033/0.034 | -0.037/0.040 |

**Table 3.** Coefficient of determination ($R^2$), root mean square error (RMSE), mean bias (MB), and number of coincident data (N) for the AOD comparison between CE318-AERONET and CE318-LUT at three different spectral bands (440, 675 and 870 nm) performed at Izaña (IZO), Santa Cruz (SCO) and Tamanrasset (TAM) stations in 2013.

| Station | Wavelength (nm) | $R^2$ | $RMSE$ | $MB$ | $N$ |
|---------|-----------------|-------|--------|------|-----|
| IZO | 870 | 0.97 | 0.011 | -0.018 | 180 |
| | 675 | 0.98 | 0.010 | -0.020 | 180 |
| | 440 | 0.97 | 0.011 | -0.018 | 180 |
| SCO | 870 | 0.99 | 0.021 | -0.021 | 210 |
| | 675 | 0.99 | 0.021 | -0.019 | 210 |
| | 440 | 0.99 | 0.021 | -0.020 | 210 |
| TAM | 870 | 0.98 | 0.030 | -0.031 | 321 |
| | 675 | 0.98 | 0.030 | -0.022 | 321 |
| | 440 | 0.97 | 0.032 | -0.023 | 321 |

**Table 4.** Coefficient of determination ($R^2$), root mean square error (RMSE), mean bias (MB), and number of coincident data (N) for the AOD comparisons between CE318-AERONET and ZEN-R41 at four different spectral bands (440, 500, 675 and 870 nm) performed at Izaña station in 2015.

| Wavelength (nm) | $R^2$ | $RMSE$ | $MB$ | $N$ |
|-----------------|-------|--------|------|-----|
| 870 | 0.97 | 0.026 | - 0.020 | 616 |
| 675 | 0.97 | 0.026 | -0.025 | 616 |
| 500 | 0.97 | 0.026 | -0.029 | 616 |
| 440 | 0.97 | 0.027 | -0.030 | 616 |

[Figure]

**Figure 1.** Meteosat/TERRA image showing a Saharan dust outbreak over the study area on 12 January 2015, where the Izaña, Santa Cruz and Tamanrasset sites are indicated with yellow stars.

[Figure]

[Figure]

**Figure 2.** ZEN-R41 scheme and dimensions.

[Figure]

**Figure 3.** AOD difference versus SZA for different AOD conditions ($AOD_{500nm} \sim 0.5$ and 1.0) and ZSR perturbed $\pm 5\%$ and $\pm 10\%$, for 440 nm (**a**, **b**) and 870 nm (**c**, **d**).

[Figure]

**Figure 4.** AOD difference versus SZA for different AOD ($AOD_{500nm} \sim 0.5$ and 1.0) and 440 nm (**a**, **b**) and 870 nm (**c**, **d**) wavelengths. We have included different concentrations of four different aerosol components, MINM, MIAM and MICM perturbed in $\pm 5\%$ and WASO concentration perturbed in $\pm 50\%$.

[Figure]

**Figure 5.** AOD scatterplot at 440 nm between CE318-AERONET and CE318-LUT for Izaña (**a**), Santa Cruz (**b**) and Tamanrasset (**c**) stations in 2013 for $20°<SZA<65°$ (**a**, **b**, **c**). The black solid lines are the least-square fits, and the dashed lines are the diagonals ($y = x$). The least-square fit parameters are shown in the legend (slope, intercept, correlation coefficient (R) and number of data (N)). AOD difference in 440 nm between CE318-AERONET and CE318-LUT respect to CE318-AERONET (**d**, **e**, **f**) and SZA (°) (**g**, **h**, **i**), respectively.

[Figure]

**Figure 6.** AOD scatterplot at 675 nm between CE318-AERONET and CE318-LUT for Izaña (**a**), Santa Cruz (**b**) and Tamanrasset (**c**) stations in 2013 for $20° < SZA < 65°$ (**a**, **b**, **c**). The black solid lines are the least-square fits, and the dashed lines are the diagonals ($y = x$). The least-square fit parameters are shown in the legend (slope, intercept, correlation coefficient (R) and number of data (N)). AOD difference in 675 nm between CE318-AERONET and CE318-LUT respect to CE318-AERONET (**d**, **e**, **f**) and SZA (°) (**g**, **h**, **i**), respectively.

[Figure]

**Figure 7.** AOD scatterplot at 870 nm between CE318-AERONET and CE318-LUT for Izaña (**a**), Santa Cruz (**b**) and Tamanrasset (**c**) stations in 2013 for 20°<SZA<65° (**a**, **b**, **c**). The black solid lines are the least-square fits, and the dashed lines are the diagonals ($y = x$). The least-square fit parameters are shown in the legend (slope, intercept, correlation coefficient (R) and number of data (N)). AOD difference in 870 nm between CE318-AERONET and CE318-LUT respect to CE318-AERONET (**d**, **e**, **f**) and SZA (°) (**g**, **h**, **i**), respectively.

[Figure]

**Figure 8.** AOD comparisons between CE318-AERONET and ZEN-R41 for four different spectral bands (440, 500, 675 and 870 nm) performed at Izaña station in 2015. In the upper panel (**a-d**) AOD scatterplots AERONET/ZEN41 are presented. The middle panel (**e-h**) shows the AOD differences versus AERONET AOD. AOD differences versus solar zenith angle (SZA in °) are shown in the lower panel (**i-l**).